# Mobile Aerosol Raman Polarizing Lidar LOSA-A2 for Atmospheric Sounding

**Sergei Nasonov [1,*] , Yurii Balin [1] , Marina Klemasheva [1] , Grigorii Kokhanenko [1] , Mikhail Novoselov [1] , Iogannes Penner [1] , Svetlana Samoilova [1] and Tamara Khodzher [2]**

[1] V.E. Zuev Institute of Atmospheric Optics, Siberian Branch, Russian Academy of Sciences, 119991 Moscow, Russia; balin@iao.ru (Y.B.); marina@iao.ru (M.K.); kokh@iao.ru (G.K.); novoselov@iao.ru (M.N.); penner@iao.ru (I.P.); ssv@seversk.tomsknet.ru (S.S.)

[2] Limnology Institute, Siberian Branch, Russian Academy of Sciences, 119991 Moscow, Russia; khodzher@lin.irk.ru

\* Correspondence: nsvtsk@gmail.com; Tel.: +7-3-822-491-283

**Abstract:** The mobile aerosol Raman polarizing lidar LOSA-A2 designed at V.E. Zuev Institute of Atmospheric Optics SB RAS is presented. Its main technical specifications are given. The lidar carries out sounding of the atmosphere of a Nd:YAG laser at two wavelengths, 1064 nm and 532 nm. Optical selection of lidar signals at these wavelengths is performed by two identical telescopes with diameters of 120 mm and a focal length of 500 mm. In the visible channel, the signal is divided into two orthogonal polarized components, and a Raman signal at a wavelength of 607 nm is separated. The lidar was tested in aircraft and ship research expeditions. Results of the study of spatial aerosol distribution over the Baikal with the use of LOSA-A2 lidar received during ship-based research expeditions are described. The first in situ tests of the lidar were carried out in an aircraft expedition in the north of Western Siberia.

**Keywords:** lidar; aerosol; atmosphere; wildfires; Lake Baikal; Western Siberia

## 1. Introduction

Studies of climate changes require reliable data on the spatial distribution and optical and physical properties of atmospheric aerosol. This information serves to increase the accuracy of calculations of radiation propagation through the atmosphere [1]. Aerosol is one of the main radioactive components in the atmosphere and largely determines its optical properties. Aerosols can be of both natural and anthropogenic origin. The vertical structure of aerosol fields is formed under regional (terrain, vegetation cover, etc.) and global factors (activity of volcanoes, sandstorms, etc.).

Remote laser sounding techniques are being actively developed worldwide. They are used in atmospheric research, first of all, for detecting air pollution sources and controlling the transboundary aerosol transfer. Lidar techniques allow studies over vast territories without sampling, at a wide range of altitudes and with high temporal and spatial resolution. The scale of the studies of optical and microphysical aerosol characteristics can vary from local one-point (with stationary or mobile devices) [2–4] to large-scale, e.g., transport of atmospheric pollutants (aircraft or satellite observations) [5,6].

The current experience of international cooperation in this field allows the creation of regional and global networks for remote, mainly optical, monitoring of the atmosphere. Thus, 20 years ago, 19 European lidar stations from 11 states were combined into the European Aerosol Research Lidar Network EARLINET for controlling the spatiotemporal distribution of atmospheric aerosol fields over Europe [7]. Later on, the Asian Dust Lidar Network AD-Net was created with similar

purposes, i.e., for the study of the transboundary transport of aerosol impurities due to sandstorms in deserts [8]; it joins scientific teams from China, Japan, and Korea. The lidar network of Commonwealth of Independent States CIS countries, CIS-LiNet [9], was created with similar aims by six scientific organizations from Russia, Belarus, and Kyrgyzstan, including V.E. Zuev Institute of Atmospheric Optics, Siberian Branch, Russian Academy of Sciences (IAO SB RAS), in 2004.

Lidar systems are key tools in these networks, which can make a significant contribution to our knowledge of atmospheric aerosols. Design of mobile small-sized lidar complexes is a promising means for further development of techniques and remote monitoring of the atmosphere. The advantage of such devices is the possibility of their mounting on mobile carriers (cars, ships, and aircraft) and use for the study of spatial distribution of aerosol fields in severe expeditionary conditions on both local and global scales. These systems must be well resistant to various mechanical loads and adverse environmental conditions.

In view of this, a mobile aerosol Raman polarizing lidar LOSA-A2 was created at IAO SB RAS in 2018 for in situ studies of surface and boundary tropospheric layers in different regions. This paper is devoted to the description of the lidar.

## 2. Design and Optical Scheme of the Lidar

Capabilities of lidars in modern studies of aerosol fields with the use of laser sounding techniques are gradually enhanced via an increase in the number of sounding radiation wavelengths and the use of channels for detecting Raman signals and polarization components of elastic backscattering signals. This allows extension of the set of retrievable aerosol optical characteristics, such as the Angstrom exponent and the color ratio, while additional polarization measurements allow more reliable identification of aerosol type [10,11].

A mobile multiwave aerosol Raman polarizing lidar LOSA-A2 was created and tested at IAO SB RAS in 2018. It is intended for remote monitoring of the atmosphere in ship-based and aircraft expeditions. The lidar is shown in Figure 1 in different options for its mounting. Up-to-date solutions were used in the design. High requirements were posed on resistance to different loads in order to keep optical adjustment during long-term transportation. In addition, the lidar should not be too large. Therefore, the LOSA-A2 case is made of high-strength carbon fiber with weak temperature dependence; it protects from the adverse weather conditions. The main transceiver components (laser, optical elements, and photo detectors) are fixed on both sides of a platform inside the case. The design of the lidar case allows mounting on a scanning platform in order to obtain spatial sections of aerosol fields in vertical and horizontal planes (for example, for mapping the spatial distribution of aerosol pollution over a city from the ground) or onboard aircraft.

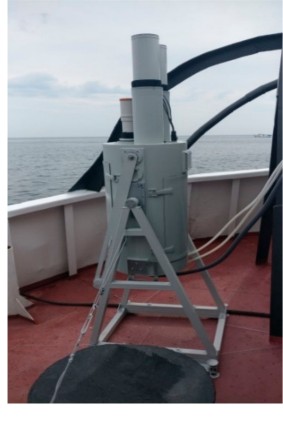 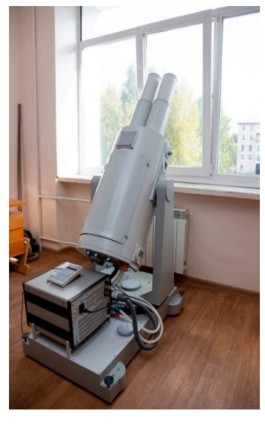 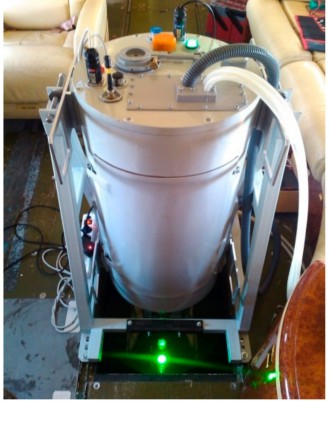

(**a**)　　　　(**b**)　　　　(**c**)

**Figure 1.** Mobile multiwave aerosol Raman polarizing lidar LOSA-A2 mounted onboard (**a**) a ship, (**b**) an automated scanning platform (stationary version), and (**c**) an aircraft.

The optical scheme of the lidar is shown in Figure 2. The system specification is given in Table 1. Sounding is carried out by radiation of a LOTIS TII Nd:YAG laser (Belarus) at two wavelengths of 1064 and 532 nm with a pulse repetition rate of up to 20 Hz. This increases the spatial resolution along the path, which is important for aircraft sounding.

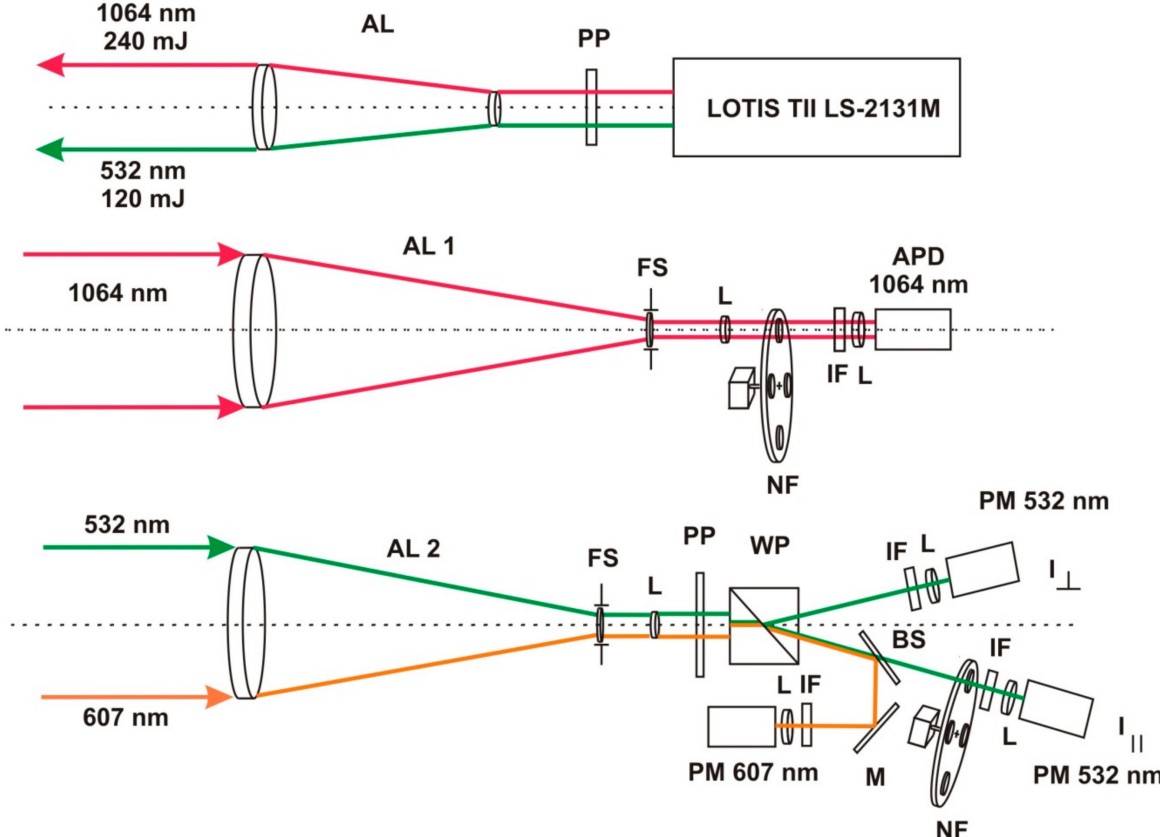

**Figure 2.** The optical scheme of the lidar: LOTIS Nd: YAG laser, polarization plates (PP), achromatic Table 1 and Al2), motorized iris diaphragms (FS) which form the field-of-view, lenses (L), motorized units of changeable neutral filters (NF), interference filters (IF), avalanche photodiode detector (APD), Wollaston prism (WP), photomultipliers (PM), dichroic interference mirror (BS), and mirror (M).

The radiation is broadened by a collimator, which allows the reduction of the beam divergence angle to 0.2 mrad. A design feature of the lidar is the use of two identical receiving telescopes with diameters of 120 mm and a focal length of 500 mm for separate recording of signals in the visible and IR spectral regions. Separate reception of backscattered radiation by two receiving lenses makes possible polarization measurements with high energy potential. Additional hoods of the telescopes are used to attenuate background noise. The heating elements built into them prevent condensation on objective lenses under adverse weather conditions.

AL1 lens is intended for receiving backscattered radiation at a wavelength of 1064 nm, where an original photo detector module developed on the basis of a C30956E-EC avalanche photodiode (PerkinElemer, Waltham, MA, USA) [12] is used. Analog signals are digitized by 12-bit ADCs LAn10-12USB-U (Rudnev–Shilyaev company, Moscow, Russian Federation) [13]. Digitalization of analog signals is performed with a minimum spatial resolution of 1.5 m.

There is a possibility of polarization measurements at a wavelength of 532 nm. A Wollaston prism is in the receiving part of this channel; it forms two mutually orthogonally polarized beams, which are simultaneously recorded by the photomultipliers (PMTs). H11526-20-NF PMTs (Hamamatsu Photonics, Hamamatsu, Japan) are used as detectors; they operate in the analog mode with controlled electronic

shutters. Unlike analogs, the shutters are closed in the normal position in the PMTs used. This allows a significant increase in the signal-to-noise ratio, since a limited time interval for signal recording substantially reduces the effect of the background solar component. Opening time of the shutter is 200 ns. The duration of a signal measurement at a sounding path is usually software-specified equal to 40 μs in the LOSA-A2 lidar.

**Table 1.** LOSA-A2 lidar specification.

| Parameter | Value |
| --- | --- |
| Transmitter | |
| Nd:YAG laser | LOTIS LS-2131M |
| Energy, mJ, at the wavelength: | |
| 1064 nm | 240 |
| 532 nm | 120 |
| Pulse length, ns | 8 |
| Pulse repetition rate, Hz | 1–20 |
| Energy stability, % | ±1 |
| Collimator | |
| Beam divergence, mrad | 0.2 |
| Beam diameter, mm | 50 |
| Output polarization | linear |
| Optical receiving system | |
| Receiving telescope | 2 |
| Telescope diameter, mm | 110 |
| Focal length, mm | 500 |
| Variable fields-of-view of the telescope, mrad | 1–10 |
| Receiving channel 532 nm | |
| Receiving mode at 532 nm | Analog (polarized, cross-polarized) |
| Detector | PMT |
| Filter bandwidth, nm | 1–2 |
| Receiving channel 1064 nm | |
| Receiving mode at 1064 nm | Analog |
| Detector | APD |
| Filter bandwidth, nm | 3 |
| Receiving channel 607 nm | |
| Receiving mode at 607 nm | Photon count |
| Detector | PMT |
| ADC | |
| Number of analog inputs | 4 |
| Input resistance, Ohm | 50 |
| Input voltage ranges, V | ±2, ±1, ±0.4, ±0.2 |
| Resolution, bit | 12 |
| Sampling frequency, MHz | 12.5, 25, 50, 100 |
| Operation conditions | |
| Temperature, °C | 0–40 |
| Humidity, % | 10–95 |

In this segment of the receiving path, a part of the radiation which arrives at the dichroic mirror (BS) is reflected by mirror M and enters the 607 nm channel, where a spontaneous Raman scattering signal from atmospheric nitrogen molecules is recorded by the H11706P-40-MOD photo detector (Hamamatsu Photonics, Japan), which operates in the photon counting mode with a controlled shutter. Then, the pulses arrive at a counter (which was designed at IAO SB RAS). This channel is used in the dark hours, in the accumulation mode. This engineering solution in the LOSA-A2 lidar allows correct retrieval of quantitative optical parameters of the atmosphere (attenuation coefficient and aerosol optical depth (AOD)) [14].

After recording, lidar data are visualized and processed in a laptop with special software complex.

Atmospheric sounding can be carried out around the clock, both in the day and night, at altitudes ranging from the surface layer to the upper troposphere (100–12,000 m), with a spatial resolution of 6 m.

The LOSA-A2 lidar is included in the scientific equipment of the "Atmosfera" Common Use Center of IAO SB RAS [15].

## 3. Calibration of Polarization Channels

Laser radiation sent to the atmosphere is linearly polarized. In the receiving part of the LOSA-A2 lidar system, the possibility of optical separation of the received signal into the parallel and cross-polarized components is realized for the 532 nm channel by means of the Wollaston prism. After passing through the Wollaston prism, signals are simultaneously recorded by two identical PMTs. The depolarization ratio is calculated as the ratio of the perpendicular component to the parallel component of the backscatter signals:

$$\delta = \frac{P_\perp}{P_\parallel} \tag{1}$$

To measure the absolute value of the backscattered radiation depolarization ratio, it is necessary to know the response ratio $V$ of the detectors. For this, the polarization calibration is carried out during experiments. The relative sensitivity of the photo detectors can be determined by measuring depolarization in free air, where the contribution of the aerosol component is negligible [16,17]. In this layer, the volume depolarization ratio is mainly determined by the molecular scattering, which is accurately theoretically calculated in [18]. This polarization calibration technique depends on the weather and state of the atmosphere and is extremely rarely implementable. The polarization calibration by procedures, where the basis polarization plane of the recording system is rotated with respect to the laser polarization plane ($\phi = 0°$) through the angle $\phi = 90°$ [19], is more widely used in lidar networks. In the LOSA-A2 lidar, this procedure is carried out by precision rotation of the platform of the 532 nm receiving channel, with photo detectors and the Wollaston prism, through the angles $\phi = \pm45°$. In this case, insignificant deviations from the mounting angles specified are additionally compensated, and the response ratio V is determined as

$$V = \sqrt{\delta_{+45°} \times \delta_{-45°}} \tag{2}$$

The response ratio of the detectors is found to be $V \approx 11$ in the experimental LOSA-A2 lidar calibration.

## 4. Results of In Situ Tests of the Lidar

The first in situ tests of the LOSA-A2 lidar took place on the expedition of 2018 to Lake Baikal. The Baikal is a unique natural object in Eastern Siberia. The great extent of the lake from the southwest to northeast, a huge mass of water, and mountainous surroundings (the height of individual mountains attains 2500 m above sea level) cause unique climatic conditions and significant differences in the distribution of meteorological parameters in the air above the lake basin. A complex system of hollow circulation of air over the Baikal [20,21] determines the formation and transport of atmospheric aerosol fields.

The anthropogenic impact on the ecosystem of Lake Baikal has been actively increasing in recent years, due to, first of all, development of tourism in the coastal zone of the lake [22–24]. About 1.5 million tourists visit the region per year. This number annually increases by 3–4% according to the regional tourism agency [25]. Another significant source of air pollution in the Baikal region is smoke aerosol emissions from wildfires, the amount of which is increasing due to the current climate warming [26]. Therefore, environmental protection measures are scheduled in the region with the aim of preserving Lake Baikal and weakening the negative impact on the environment.

Continuing studies aimed at the analysis and estimation of air pollution over Lake Baikal in summer, comprehensive experimental studies of the vertical structure of aerosol above the lake were carried out in summers 2018 and 2019 with the use of the LOSA-A2 lidar onboard the "Academician V.A. Koptyug" research vessel (RV). Their purpose was to develop physical models of the formation and transport of atmospheric aerosol fields, taking into account the physical and geographical features of the Baikal region. The preliminary results of the analysis of data received in the expedition of 2018 are presented in [27]. They show the complex dynamics of filling and formation of the altitude distribution of aerosol impurities from wildfires above the Baikal basin.

In 2019, the expedition took place from 24 July to 4 August. It continued the studies of the atmosphere over Lake Baikal. Figure 3 shows the spatial distribution of the integral backscattering coefficient β over the water area of the lake along the RV route (for a wavelength of 532 nm) [28] calculated for the lower 1-km air layer. The main aerosol content is observed in this layer and the contribution of clouds is excluded. Each point has been plotted with 20-min accumulation. The total route length was ~1700 km. The expedition began in Listvyanka on 24 July. On the first day, measurements were performed in South Baikal (Listvyanka–Baikalsk–Kultuk–Listvyanka). From 25 to 28 July, the RV northward moved along the west shore, to Nizhneangarsk, and from 28 July to 3 August, southward along the eastern shore of Lake Baikal.

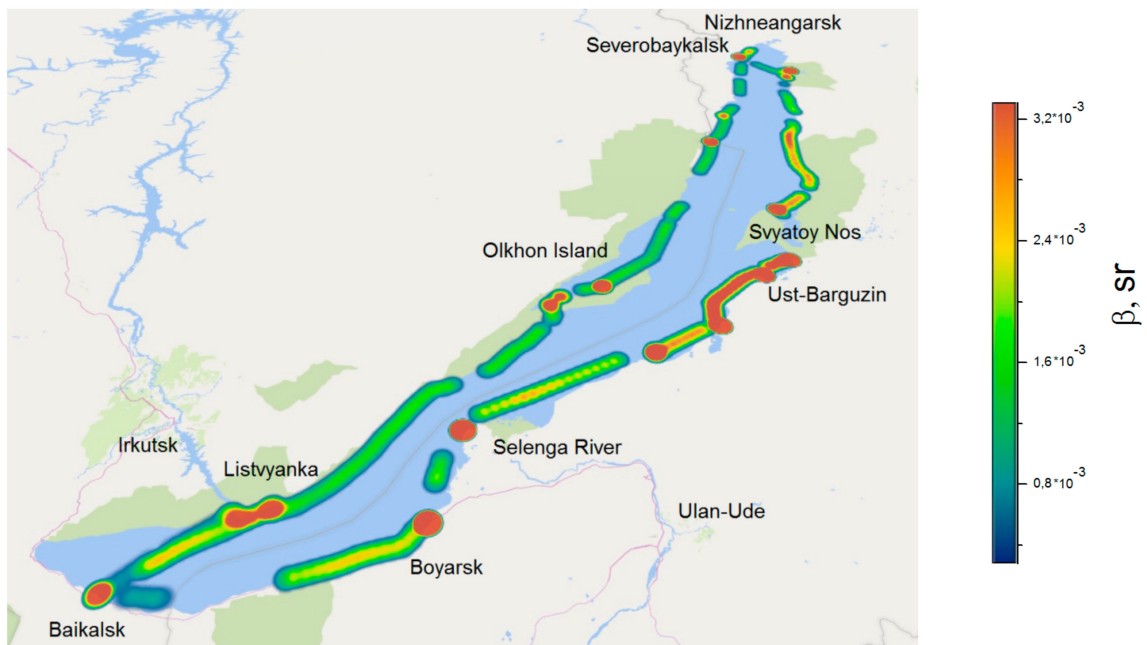

**Figure 3.** Spatial distribution of the integral backscattering coefficient for the lower 1-km air layer over Lake Baikal.

Background values of the atmospheric aerosol content were observed from 25 to 29 July, when the RV moved along the western shore in the direction from Listvyanka village to Olkhon island. Thanks to the action of a cyclone which came from the northwest, with low clouds and occasional rain, the atmosphere during this period was cleared by precipitation.

The local regions of high aerosol content as compared to the background values are shown in red. The aerosol concentration increased in the 1-km air layer mainly due to the contribution of anthropogenic sources. First of all, these are regions with developed tourism (Listvyanka and Olkhon) and industrial centers (Baikalsk and Severobaikalsk).

A section of the route along the eastern shore from north to south, from Khakusy bay to Boyarsk, should be especially noted. The maximal aerosol concentration was observed near Svyatoy Nos peninsula and Ust-Barguzin village. In this part of the expedition, a large amount of smoke aerosol

from wildfires in the Krasnoyarsk Territory and Yakutia was detected in the air above the lake. Maps of foci of wildfires in Siberia in the period under study constructed on the basis of satellite data can be found in [29].

An example of lidar signals for daytime received on the expedition of 2019 is shown in Figure 4. The signal-to-noise ratio is good enough to detect aerosol layers.

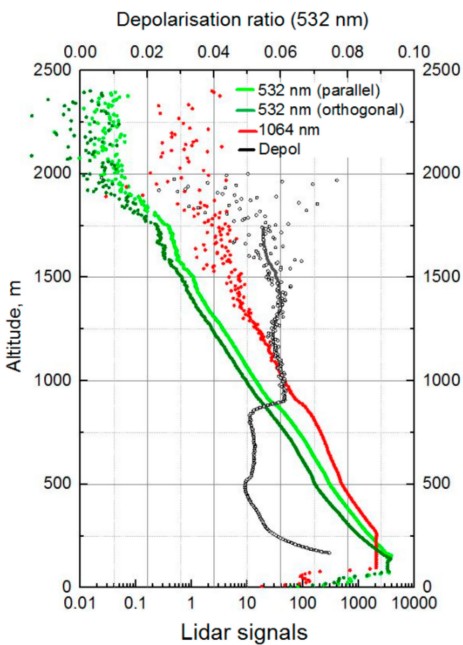

**Figure 4.** Example of vertical profiles of range-corrected lidar signals of 3 channels and depolarization ratio.

Figure 5 exemplifies the vertical cross-section of a tropospheric aerosol field retrieved from LOSA-A2 lidar data. The instant appearance of a smoke plume is seen. It was recorded on the night of 29 July 2019, at an altitude of 2–2.5 km. The color scale on the right side of the figure corresponds to the scattering ratio, i.e., the ratio of the total backscattering coefficient to the molecular backscattering coefficient [30]. In the morning of 29 July, a thick near-water morning fog overloaded the lidar photo detectors, and it had to be turned off for a while. By the evening of 29 July, the smoke aerosol began to precipitate. High scattering ratios (R~25) indicate high concentrations of aerosol particles in air. The next three days of the expedition, from 30 July to 2 August, the atmospheric situation worsened to smog. Under the influence of an anticyclone that passed over the region from the northwest to southeast, air masses from the north and west arrived, with a large amount of smoke aerosol from wildfires, which filled the air in the mountain basin with impurities.

According to the Federal Budgetary Institution "Avialesohrana", in summer 2019, when the expedition took place, the total area of wildfires in Siberia reached 3 million hectares [31]. The lidar observations showed the relative aerosol content to be approximately three times higher than the background values in the 1-km air layer from 30 July to 2 August.

An example of lidar signal for nighttime for 607 nm channel received in the expedition of 2019 and AOD calculated by the spontaneous Raman scattering SRS technique is shown in Figure 6 (before smoke plume—blue curve; during smoke plume—red curve). A maximum value of $\tau \approx 4$ was attained due to the contribution of smoke aerosol at the night of 31 July to 1 August, while the average AOD value for the background atmospheric conditions above the lake was $\tau \approx 0.1$.

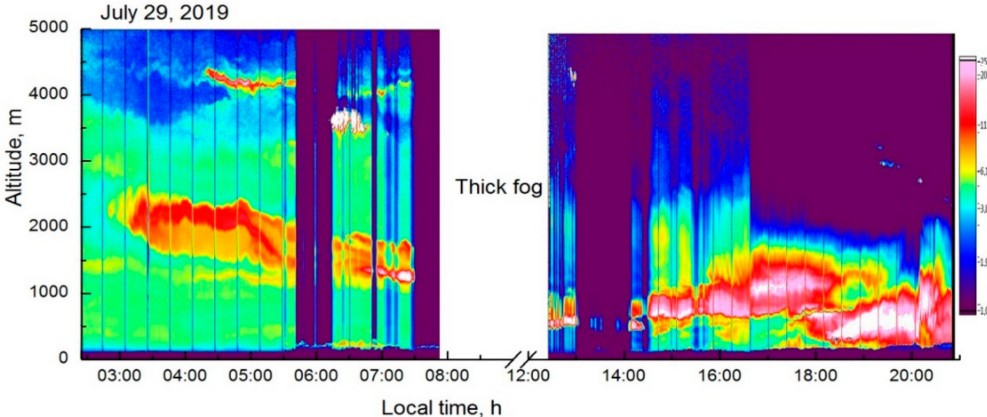

**Figure 5.** Spatiotemporal structure of the aerosol field according to data of the LOSA-A2 lidar mounted onboard "V.A. Koptyug" Research Vessel.

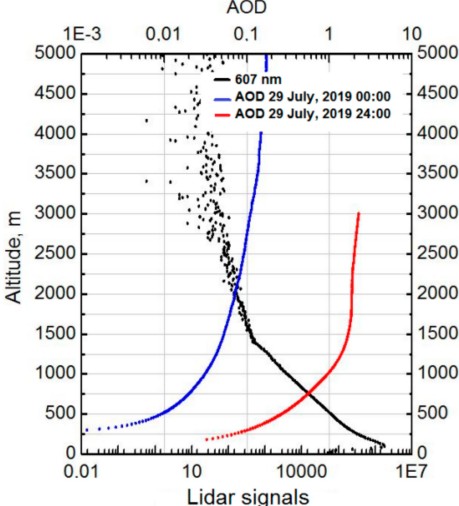

**Figure 6.** Vertical profiles of lidar signal at 607 nm and AOD on 29 July 2019: before smoke plume (blue) and during smoke plume (red).

Figure 7 allows the comparison of AOD calculated from the LOSA-A2 measurements and from the data of the global aerosol model Navy Aerosol Analysis and Prediction System (NAAPS) [32].

The figure shows good agreement between the quantitative values of the obtained data; however, the AOD calculated by the NAAPS model gives a forecast with a 1-day advance. AOD calculated from lidar data gives more accurate values.

The forecast of the propagation of aerosol fields over a mountain basin is a quite complicated task. It requires operational data on many input parameters (the presence of temperature inversions in the atmosphere, relief, direction and speed of airflows, etc.). Regular experimental studies of the atmosphere over Lake Baikal promote verification of theoretical calculations, which is to favor the further development of an adequate physical model for the formation and transport of atmospheric aerosol fields and, hence, more accurate forecast of environmental conditions in the Baikal basin. Smoke aerosol is a good tracer in such studies; it allows tracing of the dynamics of airflows over the mountain lake. This proves the urgency of the research performed and prospects of the LOSA-A2 lidar for receiving relevant information for solution of climatic problems.

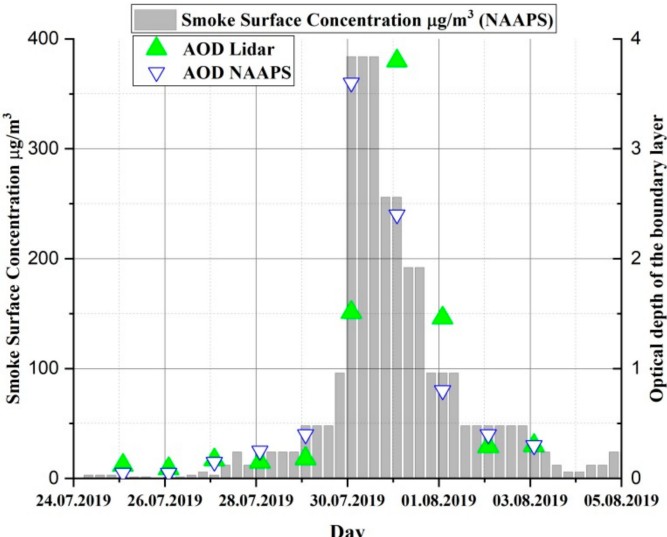

**Figure 7.** AOD calculated from the LOSA-A2 (green triangles) and NAAPS (blue triangles) data and smoke aerosol concentration, μg³ (grey) [32].

An aircraft expedition with participation of the LOSA-A2 lidar was carried out in the fall of 2018. The lidar was mounted onboard the OPTIK-E aircraft laboratory [6] and was used for vertical downward sounding through the standard photohole of the aircraft (see Figure 1c). The flight took place within the Russian–French YAK-AEROSIB project (Airborne Extensive Regional Observations in SIBeria) [33]. The aims of this project include regular experiments in the north of Western Siberia in order to study the large-scale vertical distribution of main trace atmospheric gases and aerosols and optical and meteorological parameters of air, as well as sounding of the underlying surface, including water.

The aircraft flight path Surgut–Norilsk–Igarka and the spatial structure of the aerosol field over this territory are shown in Figure 8. A plume of anthropogenic aerosol was revealed during the flight over Norilsk (this path segment is shown in magenta). High values of the scattering ratio indicate high aerosol content above the city. Norilsk is the northernmost and one of the most polluted cities on the planet. The world's largest mining and smelting plant, Norilsk Nickel, is located there; every day, it emits a huge amount of harmful substances into the atmosphere.

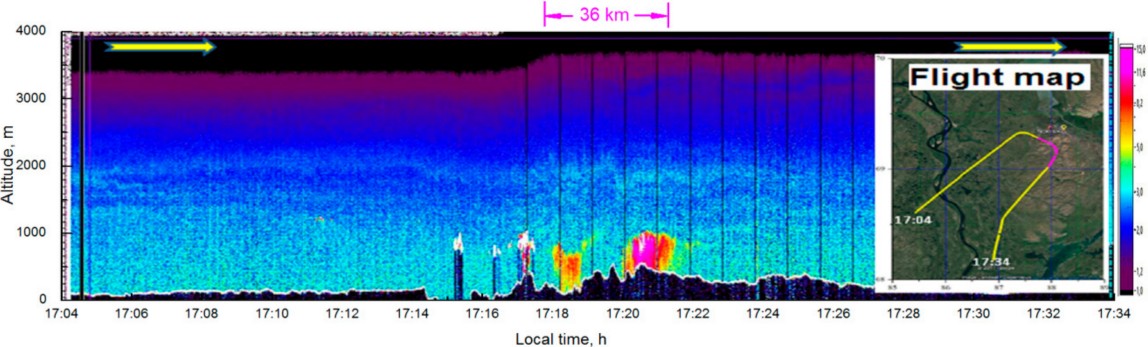

**Figure 8.** Spatiotemporal structure of the aerosol field according to data of the LOSA-A2 lidar onboard the OPTIK-E aircraft laboratory.

An example of the vertical profile lidar signal for airborne measurement is shown in Figure 9.

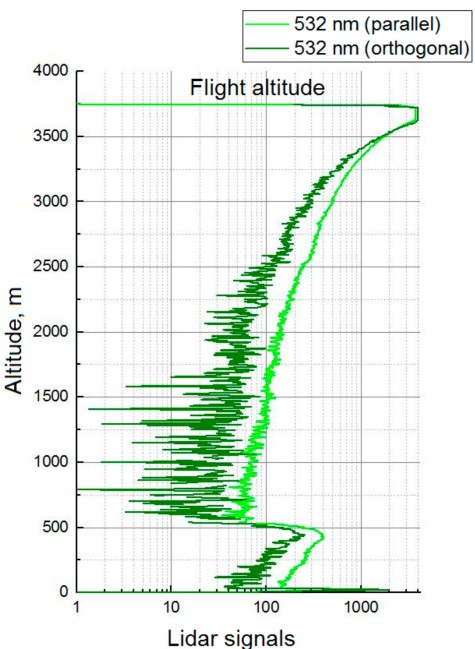

**Figure 9.** Example of vertical profiles of range-corrected lidar signal for airborne measurement.

The lidar was successfully tested for stability under conditions of strong vibrations during aircraft takeoff and landing during this expedition. The aircraft tests of the lidar showed its sufficient energy potential for environmental monitoring of the air basin of large industrial centers from aircraft.

## 5. Conclusions

The multiwave mobile aerosol Raman polarizing lidar LOSA-A2 designed by V.E. Zuev Institute of Atmospheric Optics has shown its effectiveness in the study of the atmosphere in ship-based and aircraft expeditions. The main features of the LOSA-A2 lidar are the following: (a) it is capable of simultaneously recording the elastic scattering signals in visible and infrared wavelength ranges by means of two telescopes, as well as the signals of spontaneous Raman scattering in the photon counting mode at the wavelength of 607 nm; (b) it is capable of selecting the boundaries of the necessary altitude range using the computer-controlled strobing photodetectors, that decreases the effect of noise and increases the signal-to-noise ratio; (c) it has increased structural strength of the interconnection of the optical elements of the transmitter-receiver, which is necessary for measurements in expeditionary conditions.

The spatial distribution of aerosol impurities over the water area of Lake Baikal was mapped along the ship route on the basis of lidar measurements during the expedition of 2019. The map clearly shows the main regions of high concentrations of air pollutants. Anthropogenic sources and abnormal amounts of smoke aerosol from wildfires in Siberia predictably significantly contribute to the high content of atmospheric aerosol. The resulting 2D aerosol fields made it possible to detect the initial time of the drift of a wildfire smoke plume into the air above the lake and to trace further complex processes of filling the atmospheric boundary layer due to the peculiarities of the air flow distribution over the mountainous terrain. The lidar potential for the detection of air pollutant sources from aircraft is shown.

The LOSA-A2 reliability in transportation and operation was checked, including under severe weather conditions; it provided tight coupling of all optical elements and safe operation of the system.

The lidar can be used in environmental monitoring of industrial emissions and smoke plumes, mapping urban air pollution, and studying the processes of transboundary transport of aerosol fields. The design features of the lidar allow its use both stationary and mounted on different mobile carriers.

**Author Contributions:** Y.B.: conceptualization, methodology, writing—reviewing and editing; M.K.: investigation, formal analysis; G.K.: methodology, investigation, formal analysis; S.N.: investigation, formal analysis, writing—original draft; M.N.: software, validation, data curation; I.P.: methodology, investigation, formal analysis; S.S.: methodology; T.K.: project administration, supervision, investigation. All authors have read and agreed to the published version of the manuscript.

**Funding:** The LOSA-A2 lidar was created under the financial support of the Ministry of Science and Higher Education of the Russian Federation (budget funds for IAO SB RAS). The expedition to Lake Baikal was organized and carried out under the financial support of the Russian Science Foundation (grant no. 19-17-20058).

**Conflicts of Interest:** The authors declare no conflict of interest.

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
