# Peer review of "Mobile Aerosol Raman Polarizing Lidar LOSA-A2 for Atmospheric Sounding"

_atmosphere, doi:10.3390/atmos11101032_

Round 1

Reviewer 1 Report

Please attached find my comments.

Author Response

Response to Reviewer 1 Comments:

Point 1: Please consider to include instrument characteristics in the abstract, as well as in the conclusion.

Response 1:

Abstract: The mobile aerosol Raman polarizing lidar LOSA-A2 designed at V.E. Zuev Institute of Atmospheric Optics SB RAS is presented. Its main technical specifications are given. The lidar carries out sounding of the atmosphere of a Nd:YAG laser at two wavelengths, 1064 nm and 532 nm. Optical selection of lidar signals at these wavelengths is performed by two identical telescopes with diameters of 120 mm and a focal length of 500 mm. In the visible channel, the signal is divided into two orthogonal polarized components, as well as a Raman signal at a wavelength of 607 nm is separated. The lidar was tested in aircraft and ship research expeditions. Results of the study of spatial aerosol distribution over the Baikal with the use of LOSA-A2 lidar received during ship-based research expeditions are described. The first in situ tests of the lidar were carried out in an aircraft expedition in the north of Western Siberia.

Conclusions

The multiwave mobile aerosol Raman polarizing lidar LOSA-A2 designed in Institute of Atmospheric Optics has shown its effectiveness in the study of the atmosphere in ship-based and aircraft expeditions. The main features of the LOSA-A2 lidar are the following: a) it is capable of simultaneous recording the elastic scattering signals in visible and infrared wavelength ranges by means of two telescopes, as well as the signals of spontaneous Raman scattering in the photon counting mode at the wavelength of 607 nm; b) it is capable of selecting the boundaries of the necessary altitude range using the computer-controlled strobing photodetectors, that decreases the effect of noise and increases the signal-to-noise ratio; c) it has increased structural strength of the interconnection of the optical elements of the transmitter-receiver, which is necessary for measurements in expeditionary conditions.

The spatial distribution of aerosol impurities over the water area of Lake Baikal has been mapped along the ship route on the basis of lidar measurements during the expedition of 2019. The map clearly shows the main regions of high concentrations of air pollutants. Anthropogenic sources and abnormal amounts of smoke aerosol from wildfires in Siberia predictably significantly contribute to the high content of atmospheric aerosol. The resulted 2D aerosol fields made it possible to detect the initial time of the drift of a wildfire smoke plume into the air above the lake and to trace further complex processes of filling the atmospheric boundary layer due to the peculiarities of the air flow distribution over the mountainous terrain. The lidar potential for the detection of air pollutant sources from aircraft is shown.

The LOSA-A2 reliability in transportation and operation was checked, including under severe weather conditions; it provided tight coupling of all optical elements and safe operation of the system.

The lidar can be used in environmental monitoring of industrial emissions and smoke plumes, mapping urban air pollution, and studying the processes of transboundary transport of aerosol fields. The design features of the lidar allow its use both stationary and mounted on different mobile carriers.

Point 2: Line 48: Please consider to reformulate the sentence “Therefore, these systems . . . conditions”

Response 2: These systems must be well resistant to various mechanical loads and adverse environmental conditions.

Point 3: Line 75: Please specify spatial resolution of the instrument.

Response 3: Digitalization of analog signals is performed with a minimum spatial resolution of 1.5 m.

Point 4: Line 111: What do you mean by “complex installed”? Please rephrase it.

Response 4: After recording, lidar data are visualized and processed in a laptop with special software complex.

Point 5: In Table 1 the parameters and corresponding values are hard to read. Please reorganize the table contents.

Response 5: The table has been corrected.

Point 6:  Line 213: Could you add a few sentences for discussing the AOD comparison?

Response 6: The figure shows good agreement between the quantitative values of the obtained data; however, the AOD calculated by the NAAPS model gives a forecast with a 1 day advance. AOD calculated from lidar data gives more accurate values.

Point 7:  Line 246: Please replace “us” the institute name.

Response 7: V.E. Zuev Institute of Atmospheric Optics.

Reviewer 2 Report

General comments:

The manuscript presents the results of development and testing of the mobile aerosol Raman polarizing lidar. The lidar is designed to study air pollution sources, features of aerosol layer structure, and aerosol transport over large areas. It is optimized to be installed on aboard of aircraft and ship.

The results of testing the lidar in ship and aircraft scientific expeditions across the area of Lake Baikal demonstrate the effectiveness and reliability of the developed lidar system.

The structure of the manuscript is logical. The description of the main blocks of the lidar system is detailed enough to understand the idea of the implemented technical solutions.

This paper could be helpful for the reader interested in development of lidar equipment.

The article can be published as presented  with minor technical notes:

 a part of data in the column “Value” of the table 1 is shifted down

Author Response

Point 1: a part of data in the column “Value” of the table 1 is shifted down.

Response 1: The table has been corrected.

Reviewer 3 Report

This manuscript presents a novel design of mobile two-wavelength Raman/Depolarizaion lidar system for for vehicle- or aircraft-based purpose. It can be noticed details of this system are given and some interest results from field expeditions such as transboundary transport of biomass burning are also shown. This manuscript is well organized and is valuable for researchers who want to build their own lidar system. I think this work should be published as a reference for related researches after some doubts being clarified.   Major comment:
  1. The only problem regarding to this manuscript is more technical consideration should be given. Basing on the content, I think this lidar basically is designed to monitor local or transboundary transportation of biomass burning over Siberian area on board of ship or aircraft. For example, at least the required probing range and signal-to-noise ratio for all channels should be given and discussed.
  2. According to the title this manuscript should be technique oriented. But is difficult to see the major point is to discuss the designing of lidar or to show the results of expeditions. Certainly those results are very interest. However, the authors should presents at least two or more vertical range-corrected signal profiles of the 4 channels (signal intensity vs. height, not in time-height color scale) and the product of retrievals to demonstrate the signal qualities is as good as required.
  Minor comment:
  1. Is that necessary to use two identical receiving telescopes for 1064 nm and 532 nm, respectively? In LN85-LN86, the author mentioned this design can make possible polarization measurement with high energy potential, however, I don’t think the intensity of backscatter light would not strong enough break polarization measurement. Is there any other reason? e.g. more easier to make solid structure of system to operate in hard envorinment?
  2. LN97-LN99: It mentions a shutter is used to reduce the effect of solar light. Is this shutter a mechanic shutter or optical shutter? That is quiet interest and useful information, please demonstrate the function shutter when the shutter is on and off.
  3. In section 4 (results), please show the vertical range-corrected signal profiles of each channel for selected events as mentioned. 
  4. cont. #3, if possible, I would like to see the profiles (include lidar ratio, depolarization ratio) for biomass burning, thin fog, background aerosol, and thin cloud.
  5. cont. #4, please also include at least one set of profiles for airborne measurement.

Author Response

Major comment:

  1. The only problem regarding to this manuscript is more technical consideration should be given. Basing on the content, I think this lidar basically is designed to monitor local or transboundary transportation of biomass burning over Siberian area on board of ship or aircraft. For example, at least the required probing range and signal-to-noise ratio for all channels should be given and discussed.
  2. According to the title this manuscript should be technique oriented. But is difficult to see the major point is to discuss the designing of lidar or to show the results of expeditions. Certainly those results are very interest. However, the authors should presents at least two or more vertical range-corrected signal profiles of the 4 channels (signal intensity vs. height, not in time-height color scale) and the product of retrievals to demonstrate the signal qualities is as good as required.

Response: An examples of lidar signals for daytime and night-time received in the expedition of 2019 are shown in new text.

Minor comment:

1. Is that necessary to use two identical receiving telescopes for 1064 nm and 532 nm, respectively? In LN85-LN86, the author mentioned this design can make possible polarization measurement with high energy potential, however, I don’t think the intensity of backscatter light would not strong enough break polarization measurement. Is there any other reason? e.g. more easier to make solid structure of system to operate in hard envorinment?

Response 1: This design feature makes it possible not only to carry out polarization measurements with a high energy potential, but also in the future to be able to upgrade the lidar system with additional recording channels (for example, a fluorescent channel at 680 nm).

2. LN97-LN99: It mentions a shutter is used to reduce the effect of solar light. Is this shutter a mechanic shutter or optical shutter? That is quiet interest and useful information, please demonstrate the function shutter when the shutter is on and off.

Response 2: We used in the lidar PMTs, which are ready-made industrial samples manufactured by Hamamatsu Photonics, Japan (H11526-20-NF). These PMTs work in the normally off mode and allows gate operation: 100 ns minimum gate width, 10 kHz repetition rate. This optical shutter is electronically controlled. This module also contains a high-voltage power supply so that PMT gain can be varied by simply adjusting the control voltage. These PMTs make it possible to operate in a linear mode over a wider range of recorded backscatter signals. More information about characteristics of the photodetector here: https://www.hamamatsu.com/resources/pdf/etd/m-h11526e.pdf.

  1. In section 4 (results), please show the vertical range-corrected signal profiles of each channel for selected events as mentioned.
  2. cont. #3, if possible, I would like to see the profiles (include lidar ratio, depolarization ratio) for biomass burning, thin fog, background aerosol, and thin cloud.
  3. 5. cont. #4, please also include at least one set of profiles for airborne measurement.

Response: An examples of the vertical profiles signals are shown in new text.
